# Diurnal Variation in and Optimal Time to Measure Holter-Based Late Potentials to Predict Lethal Arrhythmia after Myocardial Infarction

**DOI:** 10.3390/medicina59081460

**Published:** 2023-08-13

**Authors:** Kenichi Hashimoto, Naomi Harada, Motohiro Kimata, Yusuke Kawamura, Naoya Fujita, Akinori Sekizawa, Yosuke Ono, Yasuhiro Obuchi, Tadateru Takayama, Yuji Kasamaki, Yuji Tanaka

**Affiliations:** 1Department of General Medicine, National Defense Medical College, Tokorozawa, Saitama 359-8513, Japan; haradanaomi0612@gmail.com (N.H.); doc01083@ndmc.ac.jp (M.K.); ykawamura@ndmc.ac.jp (Y.K.); raoh0615@gmail.com (N.F.); sekiaki1@gmail.com (A.S.); onoyousuke1979@yahoo.co.jp (Y.O.); ybuchi@ndmc.ac.jp (Y.O.); yitanaka@ndmc.ac.jp (Y.T.); 2Department of General Medicine, Nihon University School of Medicine, Tokyo 173-8610, Japan; takayama.tadateru@nihon-u.ac.jp; 3Department of General Medicine, Kanazawa Medical University Himi Municipal Hospital, Toyama 953-8531, Japan; kasamaki@kanazawa-med.ac.jp

**Keywords:** late potentials, signal-averaged electrocardiography, heart rate variability, fatal arrhythmia, sudden cardiac death

## Abstract

*Background and Objectives*: Holter-based late potentials (LPs) are useful for predicting lethal arrhythmias in organic cardiac diseases. Although Holter-based LPs exhibit diurnal variation, no studies have evaluated the optimal timing of LP measurement over 24 h for predicting lethal arrhythmia that leads to sudden cardiac death. Thus, this study aimed to validate the most effective timing for Holter-based LP testing and to explore factors influencing the diurnal variability in LP parameters. *Materials and Methods*: We retrospectively analyzed 126 patients with post-myocardial infarction (MI) status and 60 control participants who underwent high-resolution Holter electrocardiography. Among the 126 post-MI patients, 23 developed sustained ventricular tachycardia (VT) (the MI-VT group), while 103 did not (the MI-non-VT group) during the observation period. Holter-based LPs were measured at 0:00, 4:00, 8:00, 12:00, 16:00, and 20:00, and heart rate variability analysis was simultaneously performed to investigate factors influencing the diurnal variability in LP parameters. *Results*: Holter-based LP parameters showed diurnal variation with significant deterioration at night and improvement during the day. Assessment at the time with the longest duration of low-amplitude signals < 40 μV in the filtered QRS complex terminus (LAS40) gave the highest receiver operating characteristics curve (area under the curve, 0.659) and the highest odds ratio (3.75; 95% confidence interval, 1.45–9.71; *p* = 0.006) for predicting VT. In the multiple regression analysis, heart rate and noise were significant factors affecting the LP parameters in the MI-VT and control groups. In the non-VT group, the LP parameters were significantly influenced by noise and parasympathetic heart rate variability parameters, such as logpNN50. *Conclusions*: For Holter-based LP measurements, the test accuracy was higher when the LP was measured at the time of the highest or worst value of LAS40. Changes in autonomic nervous system activity, including heart rate, were factors influencing diurnal variability. Increased parasympathetic activity or bradycardia may exacerbate Holter-based LP parameters.

## 1. Introduction

Since the 1990s, late potentials (LPs) detected using signal-averaged electrocardiography (SAECG) have been reported to be useful for predicting sudden cardiac death (SCD) and fatal arrhythmic events in patients with post-myocardial infarction (MI) status [1,2], dilated cardiomyopathy [3], arrhythmogenic right ventricular cardiomyopathy [4], cardiac sarcoidosis [5], and other organic heart diseases.

Recently, Holter electrocardiography (ECG) has also been used to measure LPs [6], and reports have emerged on the usefulness of Holter-based LPs for predicting SCD/lethal arrhythmia in patients with organic heart disease and chronic kidney disease [7,8,9]. Holter-based LP measurement is expected to become a mainstream test because it is performed simultaneously with routine Holter ECG, saving time for both patients and medical professionals compared to the conventional real-time LP measurement method. However, Holter-based LPs exhibit diurnal variation in post-MI patients [8], patients with Brugada syndrome [10,11], and healthy participants [12]. Given that 24 consecutive hours of LP data are obtained from Holter electrocardiographs, it is important to determine which method of LP data collection is most useful for stratifying patients by SCD/lethal arrhythmia risk. In previous studies, LP parameters, such as filtered QRS duration (fQRS) and the root mean square voltage of the terminal 40 ms in the filtered QRS complex (RMS40) captured at the time of the most abnormal or worst (lowest) RMS40 in 24 h [8,9,13] or the most abnormal or worst (highest) fQRS in 24 h [14], were used as representative Holter-based LP values. However, the timing of collecting SAECG testing data yielding LP parameters most useful for predicting lethal arrhythmias under ordinary daily conditions has not been completely validated. Moreover, factors influencing the diurnal variation in LP parameters are not completely understood in patients with post-MI status.

Therefore, this study aimed to identify the optimal timing for LP testing for stratifying the risk of post-MI patients and investigate factors influencing LP diurnal variation in post-MI patients and control participants.

## 2. Materials and Methods

### 2.1. Study Design and Ethics

In this retrospective cohort study, we initially enrolled 150 patients with post-MI status and 66 control participants, all of whom underwent high-resolution Holter electrocardiography (H-ECG) from March 2012 to December 2022 (Appendix A). Among the 150 patients with post-MI status, 33 had clinically sustained ventricular tachycardia (VT) as of March 2021 and were assigned to the MI-VT group, while 117 did not have sustained VT and were assigned to the MI-non-VT group; sustained VT was defined as ≥30 s of consecutive ventricular complexes at a rate of >100 bpm. The control group included 66 participants who underwent outpatient H-ECG for close examination of chest symptoms and ultimately showed no sign of cardiac disease.

The exclusion criteria were as follows: (1) cardiomyopathy or arrhythmogenic right ventricular cardiomyopathy, (2) persistent atrial fibrillation or flutter, (3) right or left bundle branch block and intraventricular conduction delay, (4) permanent pacing with pacemaker or implantable cardioverter defibrillator, (5) atrioventricular block Ⅱ–Ⅲ degree, and (6) channelopathies, such as long QT syndrome, Brugada syndrome, and early repolarization syndrome; 24 patients were excluded for these reasons. Finally, 126 patients with post-MI status (MI-VT group, *n* = 23; MI-non-VT group, *n* = 103) and 60 control participants were included in the study (Appendix A, Table 1).

The study protocol conformed to the Declaration of Helsinki and was approved by the Medical Ethics Committee of the National Defense Medical College Hospital (approval no. 4692), Saitama, Japan, and the Nihon University School of Medicine, Itabashi Hospital, Tokyo, Japan (approval no. MF-2302-0063).

### 2.2. Ambulatory ECG Recordings

Patients underwent H-ECG recording during ordinary daily activities at least 3 weeks after MI onset to avoid acute phase electrical instability. Data obtained from the H-ECG system (SpiderView; ELA Medical, Paris, France) were analyzed for routine arrhythmic events. The length of the H-ECG recording conducted for each patient was 24 h.

### 2.3. Measurement of Holter-Based LPs

Orthogonal X, Y, and Z bipolar leads with silver–silver chloride electrodes (Blue SENSOR^®^; METS, Tokyo, Japan) were used for all LP recordings. LPs were recorded for all patients and control participants using the H-ECG system. ECG data were obtained at a sampling rate of 1000 Hz using 16-bit A/D conversion. For the LP measurements, the ECG data were filtered and ranged from 40 to 250 Hz [6,8]. Then, the LP signals of 250 complexes were averaged (default setting). The LP parameters were automatically measured by the software during the 24 h time period; the parameter assessments were manually edited by expert electrophysiological investigators using the Syne Scope (SORIN GROUP, Milano, Italy). The expert electrophysiological investigators were blinded to patient outcomes. The LP parameters were assessed independently by two expert electrophysiological investigators, and disagreements were resolved by consensus.

Three LP parameters were evaluated in the 24 h records of the MI-VT, MI-non-VT, and control groups: the filtered QRS duration (fQRS) (ms), the root mean square voltage of the terminal 40 ms in the filtered QRS complex (RMS40) (µV), and the duration of low-amplitude signals < 40 μV in the filtered QRS complex terminus (LAS40) (ms). We evaluated each of the LP parameters every 4 h at 6 time points: 0:00, 4:00, 8:00, 12:00, 16:00, and 20:00. To adjust the LP signal for noise level, a Holter-based LP measured at 0.8 µV or less was used; however, if no part of the LP measurement had a noise level below 0.8 µV, the LP parameters were measured over a wider range of up to 2 h before and after the time period to find such a portion, and noise reduction was performed. Then, the signals at times corresponding to the (a) worst fQRS, (b) best fQRS, (c) worst RMS40, (d) best RMS40, (e) worst LAS40, and (f) best LAS40 values relative to the presence of LPs and (g) the 24 h mean values of each of the three LP parameters were selected for evaluation. The averaged signal obtained at each selected time was judged to be positive or negative for the presence of LPs (Figure 1). LPs were considered to be present when any two of the following three criteria were met: fQRS > 114 ms, RMS40 < 20 μV, and LAS40 > 38 ms [15]. Furthermore, the periodical times (0:00, 4:00, 8:00, 12:00, 16:00, and 20:00) were evaluated as representative times for assessing the presence or absence of LPs. The LP parameters of the signals at these times were judged as LP positive or negative. The body position at the time of LP measurement was estimated manually from the behavior record card.

### 2.4. Heart Rate Variability Analysis

Heart rate (HR) variability (HRV) analysis was also performed to evaluate autonomic nervous activity using the SpiderView (Ela, Paris, France) at the same time as when the Holter-based LPs were measured. HRV time- and frequency-domain analyses were conducted at 5 min intervals. For the frequency-domain analysis, the RR interval was calculated using fast Fourier transformation. The time-domain analysis included the percent difference between adjacent normal NN intervals greater than 50 ms (pNN50), the root mean squared successive differences of NN intervals (RMSSD), the mean of 5 min standard deviations of NN intervals (ASDNN) (ms), and the standard deviation of the average NN interval for each 5 min segment (SDANN) (ms). For the frequency-domain analysis, the power in the low-frequency area (LF), power in the high-frequency area (HF), and power in the low-frequency area/power in the high-frequency area (LF/HF) ratio were also analyzed every 5 min. The power spectra of the frequency-domain analysis were defined as follows: total power (TP), <0.4 Hz; power in the very low-frequency range (VLF), 0.0033–0.04 Hz; power in the low-frequency range (LF), 0.04–0.15 Hz; and power in the HF, 0.15–0.40 Hz. Based on a previously published report [16], LF normalized unit (LFnu) and HF normalized unit (HFnu) were calculated using the following formulas: LFnu = [LF/(TP-VLF)] × 100 and HFnu = [HF/(TP-VLF)] × 100. The HRV parameters were evaluated simultaneously with the LP parameters whenever the signal had an acceptable noise level <0.8 µV.

### 2.5. Statistical Analyses

The data are presented as the mean ± standard deviation for normally distributed continuous variables and as medians (interquartile range: 25th–75th percentile) for non-normally distributed variables. The patient characteristics were compared using the χ2 test for categorical variables, Student’s t-test for continuous and parametric data, and Mann–Whitney U test for nonparametric data. The distribution of the continuous variables was evaluated for normality by the Shapiro–Wilk test. Friedman’s analysis of variance (ANOVA) on rank was used to compare the LP parameters (fQRS, RMS40, and LAS40) for each LP measurement time. The sensitivity, specificity, positive predictive value (PPV), and negative predictive value (NPV) were calculated using standard formulas. A receiver operating characteristic (ROC) curve was generated, and the area under the curve (AUC) was calculated to determine the LP measurement timing that best predicted VT. Logistic regression analysis was performed to correlate the occurrence of VT with each LP measurement time (when fQRS, RMS40, and LAS40 were the worst; when they were the best; and at 0:00, 4:00, 8:00, 12:00, 16:00, and 20:00). Cochran’s Q test was performed to compare the LP positivity rates at each time point (0:00, 4:00, 8:00, 12:00, 16:00, and 20:00). Multivariate regression analysis was performed to determine the intensity of the diurnal variation in the LP parameters and theoretically consider important factors, such as the HR and HRV indices. Because the HRV indices (RMSSD, ASDNN, SDANN, pNN50, LFnu, VLF, HFnu, and LF/HF) showed skewed distributions, they were natural log-transformed before multiple regression analysis to explore factors influencing the diurnal variation in Holter-based LPs was performed.

Sample size calculation was performed based on the correlation among six repeated-measures ANOVA using R (4.2.3.2 Ver.) (R Foundation for Statistical Computing, Vienna, Austria), a two-tailed hypothesis, an effect size of 0.40, an α error probability of 0.05 with a β level of 10%, between-group variance = 5, within-group variance = 30, and a desired power analysis of 90% (1-β error probability). This calculation showed that a total sample size of at least 20 participants with sustained VT (SVT) was required to achieve the desired power. Consequently, a total of 150 patients with MI were included (120 MI-non-SVT participants) to enroll consecutive cases with at least 20 SVT. Ultimately, a total of 126 patients with MI were included based on the inclusion criteria (MI-VT, *n* = 23; MI-non-VT, *n* = 103). All statistical analyses, except for the sample size analysis, were performed using SPSS version 28 (IBM Corp., Armonk, NY, USA). The T-tests were two-sided, and p values of <0.05 were considered statistically significant.

## 3. Results

### 3.1. Patient Demographics

The demographic data, including age, sex, comorbidities, echocardiographic data, renal function parameters, and medication therapy, were extracted from the electronic medical records of the patients. Table 1 shows the characteristics of the patients and control participants included in the study. The number of patients with diabetes mellitus, culprit coronary lesions of the left anterior descending artery, left ventricular end-diastolic diameter, and amiodarone use post-MI were significantly higher in the MI-VT group than in the MI-non-VT group. Consequently, LVEF was significantly lower in the MI-VT group than in the MI-non-VT group.

### 3.2. Optimal Measurement Timing for Assessment of Holter-Based LPs

In the MI-VT, MI-non-VT, and control groups, Holter-based LPs showed significant diurnal variation for all three parameters (fQRS, RMS40, and LAS40) (Table 2). In all groups, LPs deteriorated during the nighttime (20:00–8:00) and improved during the daytime (8:00–20:00) (Table 2). The LP positivity rates of the three groups (MI-VT, MI-non-VT, and control groups) were significantly higher at night and lower during the daytime (p=0.002–0.009) (Table 3). 

MI-non-VT, myocardial infarction without ventricular tachycardia; MI-VT group, myocardial infarction with ventricular tachycardia; LP, late potential.

Table 4 shows the predictive values associated with Holter-based LPs for each parameter and time point. For each LP parameter, the NPV was not different among the LP parameters (85–89%). However, the PPV in the worst fQRS (61%), RMS40 (61%), and LAS40 (65%) tended to be better than the other PPVs of the LP parameters (43–48%) for each time setting (Table 4, right). Although the NPV at 16:00 was the lowest, the PPVs at 0:00, 4:00, 16:00, and 20:00 (57–61%) tended to be better than those during the daytime (8:00, 12:00) (43–52%).

In the ROC curve for each LP parameter, when the timing of the worst LAS40 reading was selected as the standard, the AUC was higher (0.659) and the test accuracy was the highest (Figure 2a). However, in the ROC curve for each time period (0:00, 4:00, 8:00, 12:00, 16:00, and 20:00), when 20:00 was selected as the standard for the LP measurement timing, the AUC was the highest (AUC = 0.678) and the SAECG test was highly accurate (Figure 2b). In the logistic multivariate regression analysis, the highest odds ratio was observed when LAS40 worst timing was used as the standard (odds ratio = 3.75, 95% confidence interval [CI] = 1.45–9.71, *p* = 0.006) (Table 5, “For each LP parameter” section). In contrast, the highest odds ratio was observed when 20:00 was selected as the standard for LP measurement timing (odds ratio = 4.89, 95% CI = 1.88–12.7, *p* = 0.006) (Table 5, “For each time point” section).

### 3.3. Factors Influencing Diurnal Variability of Holter-Based LPs

Regarding the factors influencing the diurnal variation in the LP parameters, multiple regression analysis revealed HR to be the factor that influenced the diurnal variation in the LP parameters in the MI-VT group the most (Table 6A). In the MI-non-VT group, the LP parameters were significantly influenced by noise and by the HRV markers of parasympathetic nervous activity, such as logpNN50 and log ASDNN (Table 6B). In the control group, the LP parameters were significantly influenced by noise and HR, pNN50 and logHFnu (Table 6C).

## 4. Discussion

This study found that Holter-based LPs and their associated factors had significant diurnal variation in the MI-VT, MI-non-VT, and control groups (Table 2 and Table 3). The optimal times for measuring Holter-based LPs to predict VT were the time of the worst LAS40 reading and nighttime (20:00) (Figure 2, Table 4 and Table 5). The factors that influenced LP diurnal variation were aspects of autonomic nerve activity, such as log pNN50 (%), logHFnu, log ASDNN, and HR (Table 6). On the other hand, noise was the factor that influenced the measurement of diurnal variation in LPs.

### 4.1. Diurnal Variation in Holter-Based LPs

It is very important that the LP itself has diurnal variation, especially in the MI-VT group (Table 2 and Table 3), because the rate of LPs assessed as positive can change according to the timing of the LP measurement. The maximum difference in the LP positivity rate between nighttime and daytime in the MI-VT group was 14% (57% at 0:00 and 43% at 8:00). Therefore, exploration of the optimal timing of Holter-based LP measurement is valuable. It has been reported that LPs vary diurnally in patients with MI [8,17,18] and healthy participants [12]. In both post-MI and healthy patients, the late potentials worsened at night and improved during the daytime. An important finding in our study is that each LP parameter in the MI-VT group showed diurnal variation around the cutoff values of the SAECG diagnostic criteria [15] (i.e., the mean values of fQRS, RMS40, and LAS40 were near 114 ms, 20 μV, and 38 ms, respectively) (Table 2). Therefore, it is key to consider diurnal variation as it relates to positive/negative LP determination. Factors reported to influence diurnal variation include HR [19], autonomic nervous system activity [20], body position [21], and physical activity [17,22]. Goldberger et al. [20] studied the effects of tilt, epinephrine, isoproterenol, beta-blockers, beta-blockers +atropine, and phenylephrine on the LP in 14 healthy participants. The results showed that the LP parameters improved with tilt-up and isoproterenol and worsened with epinephrine. Atrial pacing and atropine did not significantly change the LP parameters compared to those observed in the healthy participants. In these results, the response to tilt is of interest. Thus, endogenous sympathetic hyperactivity and decreased parasympathetic activity may be involved in the improvement of LP parameters.

### 4.2. Optimal LP Measurement Timing for Predicting VT

When the time of the worst LAS40 reading or nighttime (20:00) was used as the standard time of LP measurement over 24 h, the odds ratio for VT and the accuracy of the SAECG test were higher than for other Holter-based LP measurement times. The times of the worst fQRS and RMS40 readings were also candidates for standard test times, although they were inferior in terms of the odds ratio but not in terms of sensitivity, specificity, NPV, or PPV. Amino et al. [14] reported the usefulness of Holter-based LP assessment as a predictor of rehospitalization in patients with post-MI status. They reported that a positive LP at the time of the worst fQRS value was significantly predictive of rehospitalization. In contrast, in the multicenter collaborative study (Janan Noninvasive Risk Stratification (JANIES) study) [8], when a comparison was made between the predictive value of the readings at the times of the RMS40 worst and best levels, it was concluded that the LPs measured at the time of the worst RMS40 were more useful, with a higher risk hazard of 8.2 (*p* = 0.003) for fatal arrhythmias in patients with MI. Our study compared the usefulness of all possible patterns for the first time. In direct terms, we compared the usefulness of readings at the times of the worst and best values of fQRS, RMS40, and LAS40, and of all mean values (Figure 1).

Table 7 shows the predictive values of late potential events after MI as reported in the previous literature [23,24,25,26,27]. Generally, LP testing is characterized by high NPVs and low PPVs. Representative real-time LPs in previous studies yielded PPVs of 15–38% and NPVs of 89–91%. Of note, the PPVs found in our study (57 and 65%) were higher than those in previous studies, including those using standard real-time LP recording (Table 7). This may represent an advantage of Holter-based LPs, provided that the measurement point of LP determination is optimal. In contrast, the sensitivity was equivalent but the specificity and NPV were relatively low in our study compared to those in the previous studies. We speculate that the reason for this phenomenon is that the proportion of patients with cardiac events relative to the total was higher in this study than that in other studies (18%). Therefore, the low number of LP(-), VT(-) patients may have led to the somewhat lower specificity and NPV.

There have also been no comparative studies of optimal LP measurements according to the time of day. In general, the times of the worst fQRS, RMS40, and LAS40 readings often coincide; however, this is not always the case. The determination that the optimal timing for Holter-based LP measurement was when LAS40 was at its worst point was a novel finding. Although the LP is an automatic measurement for some electrocardiography models, others require manual editing for Holter-based LP measurements, in which case, fQRS and RMS40 measurements can be influenced by the manual editing of the onset and offset settings at the time of the LP measurement, and this may cause bias among examiners. However, the LAS40 assessment was not affected by the manual editing of the onset or offset settings. Therefore, using LAS40 to measure LP could help avoid bias among examiners.

### 4.3. Factors Influencing Holter-Based LP Values

In the multiple regression analysis, HR was the factor most influencing diurnal variation in the LP parameters in the MI-VT group (fQRS, β = 0.180, *p* = 0.037; RMS40, β = 0.305, *p* = 0.003; LAS40, β = −0.261, *p* = 0.011) (Table 6A, MI-VT group). In the MI-non-VT group, the LP parameters were significantly influenced by noise or parasympathetic activity HRV parameters, such as log pNN50, considered to reflect parasympathetic nervous activity, and logLF/HF, which is considered to reflect the balance among sympathetic, nervous, and parasympathetic nervous activity (Table 6B, MI-non-VT group). These results suggest the involvement of the autonomic nervous system in both the MI-VT and MI-non-VT groups. In addition, logASDNN was a significant factor in the MI-non-VT group. In contrast, in the control group, diurnal variability in the LP parameters was significantly influenced by log HFnu, log pNN50, or HR (Table 6C, control group). Log noise is one of the significant factors that influenced the measured diurnal variation in the Holter-based LPs in the MI-non-VT and control groups. In the multivariate analysis, noise significantly affected the LP parameters in the MI-non-VT and control groups, but not in the MI-VT group. Although it is difficult to explain the substantive reason for this difference, it is possible that the MI-VT group results were less affected by noise because of their significantly lower EF and greater LVDD compared to the MI-non-VT and control groups. Although HR did not survive in the multivariate analysis as a significant factor in the MI-non-VT group, HR is normally influenced by the autonomic nervous system. In the MI-non-VT group, log HR was mildly correlated with log pNN50 and log ASDNN, both at R = 0.3 (Pearson correlation coefficient), *p* < 0.001. Thus, in all groups, increasing parasympathetic tone or bradycardia may lead to an exacerbation of LP parameters.

Askin et al. [27] examined the characteristics of HRV in participants with no apparent cardiac disease divided into groups with more and fewer PVCs. They found that LF/HF was higher in the high PVC group than in the low PVC group both at night and during the day and that higher LF/HF was a risk factor for increased PVCs in multivariate analysis. Although the LF/HF ratio is often interpreted as a balance between sympathetic and parasympathetic activity, the authors hypothesize that sympathetic hyperactivity rather than altered parasympathetic activity underlies the PVC increase. PVCs and nonsustained VT are important because they are key triggers in the generation of lethal arrhythmias [28]. Fortunately, in our study, parasympathetic tone resulted in increased arrhythmogenicity, the opposite of the autonomic modulation in increased PVCs.

The LPs were also influenced by HR, especially in the MI-VT and control groups after 24 h. Among the LP parameters, fQRS and LAS40 are considered to be more influenced by HR than by RMS40 because bradycardia directly prolongs fQRS and LAS40. Yoshioka et al. [2] examined patients with Brugada syndrome and healthy participants and reported that the LP parameters were influenced by body position. In our results, body position was not a significant influential parameter for the LP parameters in any of the groups in the multivariate analysis (Table 6). However, this is the first demonstration of diurnal variations of LP parameters, including those related to body position. Our results indicate that body position was not a significant factor influencing the LP parameter values compared to autonomic activity, HR, and noise level in natural daily activities.

Finally, Holter-based LP analysis requires the extraction of useful data from a large dataset. Therefore, its use to predict risk is labor-intensive and imposes a large time burden on cardiologists and technicians. Therefore, artificial intelligence (AI) support will be essential for future applications. Recently, with advancements in AI technology, the accuracy of ECG analysis at the μV level has improved [29]. Based on the data from the present study, it is possible to develop AI-supported clinical practices. We believe that this will lead to the widespread use of Holter-based LP assessment.

### 4.4. Clinical Implications

Holter-based LP can be performed simultaneously with routine Holter ECG, saving time for both patients and medical staff compared to the conventional real-time LP measurement. Furthermore, according to the present study results, the likelihood of missing a good accuracy test result of Holter-based LP would be greatly decreased by reading mainly at the time worst value of LAS40 or around 20:00.

### 4.5. Limitations

This study had some limitations. Firstly, the H-ECG recorder used (SpiderView^®^) did not have an accelerometer. Although the consistency of body position was confirmed to some extent using the activity record card, this assessment was not precise, and information on body position in this study may not necessarily be accurate. Secondly, the study results only provide data for the risk stratification of fatal arrhythmias in patients with post-MI status. Therefore, it should be applied clinically with caution as it may not be indicated in other cardiac diseases, such as Brugada syndrome, ARVC, dilated cardiomyopathy, or heart failure.

## 5. Conclusions

In Holter-based LP measurement, when the time of the worst LAS40 value or nighttime (20:00) was used as the standard value for predicting VT, both the odds ratio and accuracy of the SAECG test were the highest in patients with MI. In contrast, time points taken when the fQRS and RMS40 were at their worst were also candidates as LP measurement time points. The involvement of the autonomic nervous system, reflected by HR, has been suggested as a factor influencing both patients with post-MI status and healthy control participants. Increasing parasympathetic activity or bradycardia tends to exacerbate Holter-based LP parameters.

## Figures and Tables

**Figure 1 medicina-59-01460-f001:**
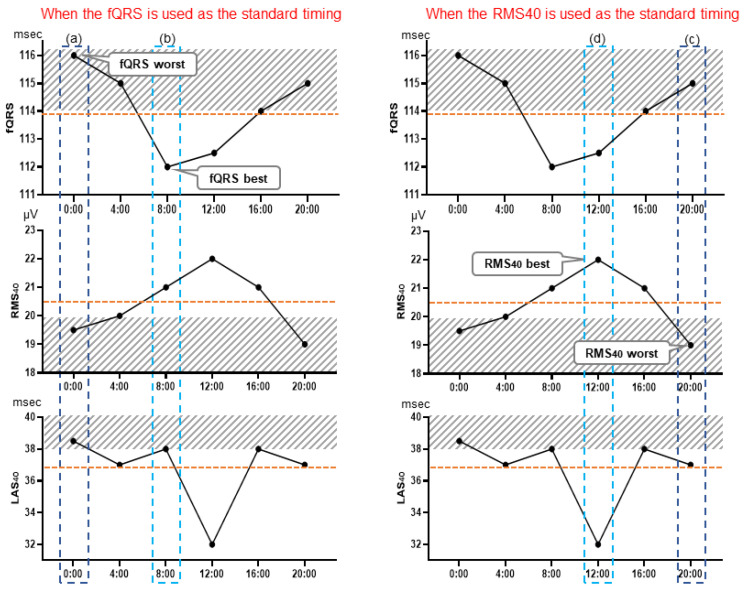
Holter-based late potential measurement point examples. These graphs illustrate the selection of measurement points: at the time of the worst fQRS point, where fQRS = 116 ms, RMS40 = 19.5 μV, and LAS40 = 38.5 ms (**a**), and at the time of the fQRS best point, where fQRS = 112 ms, RMS40 = 21 μV, and LAS40 = 38 ms (**b**). Similarly, LP determination was performed using the values of each parameter at the time of the worst (**c**) and best (**d**) RMS40 point, and of the worst (**e**) and best (**f**) LAS40 points. LP measurement was also performed using the mean value of each parameter, which were fQRS = 113.8 ms, RMS40 = 20.5 μV, and LAS40 = 36.8 ms. fQRS, filtered QRS duration; RMS40 root mean square voltage of the terminal 40 ms of the filtered QRS complex (µV); LAS40, duration of low-amplitude signals < 40 μV in the filtered QRS complex terminus (ms).

**Figure 2 medicina-59-01460-f002:**
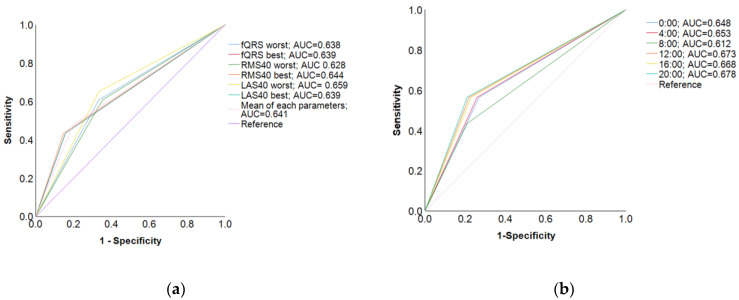
Receiver operating characteristic curves. (**a**) In the ROC curve for each parameter, when the LP parameter value was at the point with the worst LAS40, the AUC was higher (AUC = 0.659) and the test accuracy was lower; (**b**) in contrast, in the ROC curve for each time period, when the LP parameter value was at the 20:00 time point, the AUC was the highest (AUC = 0.678), and the test was highly accurate.

**Table 1 medicina-59-01460-t001:** Baseline characteristics of the study participants by group.

Demographics	MI-VT Group (*n* = 23)	MI-Non-VT Group (*n* = 103)	*p* Value	Control Group (*n* = 60)
Age (years)	66.9 ± 12.4	66.9 ± 13.1	0.994	56.7 ± 20.5
Sex: male, *n* (%)	22 (96)	83 (81)	0.195	33 (55)
Hypertension, *n* (%)	18 (78)	87 (84)	0.758	―
Dyslipidemia, *n* (%)	14 (61)	68 (66)	0.831	―
Diabetes mellitus, *n* (%)	17 (74)	41 (40)	0.002	―
Coronary culprit lesion				
RCA	3 (13)	39 (38)	0.023	―
LAD	17 (73)	43 (42)	0.04	―
Cx	2 (13)	10 (20)	0.562	―
Echocardiographic data				
LVEF (%)	48.5 ± 16.0	58.4 ± 11.9	<0.001	70.8 ± 6.5
LVDd (mm)	57.1 ± 11.6	50.1 ± 7.4	<0.001	44.4 ± 4.6
Renal function				
Estimated GFR (mL/min per 1.73 m^2^)	46.9 [34.7, 68.5]	61.3 [37.7, 76.1]	0.146	78.5 ± 18.2
Creatinine (mg/dL)	1.1 [0.8, 1.5]	0.93 [0.7, 1.2]	0.152	0.69 [0.63, 0.79]
Therapy				
β-Blocker (%)	19 (83)	77 (75)	0.424	―
RAS inhibitor (%)	14 (61)	66 (64)	0.729	―
CCB (%)	11(48)	32 (31)	0.125	―
Diuretic (%)	12 (52)	42 (41)	0.581	―
Amiodarone (%)	8 (34)	6 (6)	<0.001	―
Ⅰb (%)	1 (4.3)	5 (4.8)	0.918	―
Ⅰc (%)	0 (0)	0 (0)	―	―

Data are given as *n* (%) or mean ± SD. CCB, calcium channel blockers; Cx, circumflex branch; GFR, glomerular filtration rate; LAD, left anterior descending; LVEF, left ventricular ejection fraction; LVDd, left ventricular dimension diameter; MI-non-VT, myocardial infarction without ventricular tachycardia; MI-VT, myocardial infarction with ventricular tachycardia; RAS, renin–angiotensin system; RCA, right coronary artery.

**Table 2 medicina-59-01460-t002:** Comparison of LP parameters among the six measurement times.

		0:00	4:00	8:00	12:00	16:00	20:00	*p* Value	
MI-VT group (*n* = 23)									
fQRS (ms)	median	115.0	116.0	116.0	116.0	114.0	118.0	0.005	
	(interquartile range)	[108.0,134.8]	[108.0, 131.0]	[101.0, 135.0]	[102.0, 135.0]	[107.0, 132.0]	[107.0, 134.0]		
RMS40 (µV)	median	14.0	14.0	21.0	18.0	16.0	16.0	0.04	
	(interquartile range)	[10.3, 54.8]	[10.0, 43.0]	[11.0, 55.0]	[8.0, 57.0]	[9.0, 43.0]	[6.6, 52.0]		
LAS40 (ms)	median	43.5	41.0	37.0	40.0	40.0	39.0	0.02	
	(interquartile range)	[29.0, 53.0]	[31.0, 48.0]	[27.0, 46.0]	[27.0, 46.0]	[26.0, 51.0]	[30.0, 50.0]		
MI-non-VT group (*n* = 103)									
fQRS (ms)	median	101.0	102.5	100.5	98.0	99.0	99.0	<0.001	
	(interquartile range)	[93.0, 115.0]	[94.0, 113.5]	[91.8, 112.3]	[93.0, 114.0]	[90.0, 110.5]	[94.0, 113.5]		
RMS40 (µV)	median	30.5	30.5	32.5	34.0	36.0	30.0	<0.001	
	(interquartile range)	[16.0, 45.8]	[16.0, 45.8]	[20.0, 48.3]	[18.5, 47.0]	[19.5, 50.5]	[20.8, 48.5]		
LAS40 (ms)	median	30.0	32.0	30.0	30.0	29.0	31.0	0.03	
	(interquartile range)	24.0, 41.5]	[24.0, 39.5]	[24.0, 36.5]	[24.0, 36.5]	[24.5, 36.0]	[25.0, 36.0]		
Control group (*n* = 60)									
fQRS (ms)	median	90.0	90.0	87.5	85.0	87.0	88.0		
	(interquartile range)	[86.0, 95.3]	[87.0, 96.0]	[83.0, 93.3]	[83.8, 90.0]	[83.0, 91.0]	[83.0, 93.0]	<0.001	
RMS40 (µV)	median	45.5	44.5	49.5	55.5	53.0	47.0		
	(interquartile range)	[29.5,64.0]	[28.8, 65.8]	[31.0, 81.8]	[33.0, 81.5]	[38.3, 78.8]	[33.0, 79.6]	<0.001	
LAS40 (ms)	median	28.0	27.0	27.0	26.0	26.0	25.0		
	(interquartile range)	[23.0, 32.0]	[24.0, 31.3]	[21.0, 33.0]	[20.0, 30.3]	[21.0, 29.0]	[22.0, 31.3]	0.03	

fQRS, filtered QRS duration; LAS40, duration of low-amplitude signals < 40 μV in the filtered QRS complex terminus (ms); LP, late potential; MI-non-VT, myocardial infarction without ventricular tachycardia, MI-VT, myocardial infarction with ventricular tachycardia, RMS40, root mean square voltage of the terminal 40 ms in the filtered QRS complex (µV).

**Table 3 medicina-59-01460-t003:** Diurnal variation in the LP positivity rate in each group.

MI-VT Group (*n* = 23)							
	0:00	4:00	8:00	12:00	16:00	20:00	*p* Value
Number of patients	13 (57)	13 (57)	10 ^§^ (43)	11^#^ (48)	13 (57)	12 (52)	0.009
(%)							
MI-non-VT group (*n* = 103)							
	0:00	4:00	8:00	12:00	16:00	20:00	*p* value
Number of patients	24 (23)	23 (22)	18 ^§^ (17)	19 ^§^ (18)	21 (20)	21 (20)	0.002
(%)							
Control group (*n* = 60)							
	0:00	4:00	8:00	12:00	16:00	20:00	*p* value
Number of participants	7 (12)	4 ^#^ (7)	4 ^#^ (7)	3 ^§^ (5)	2 ^§^ (3)	3 ^§^ (5)	0.009
(%)							

^#^ = 0.005 vs. 0:00; ^§^ ≤0.001 vs. 0:00.

**Table 4 medicina-59-01460-t004:** Predictive values associated with Holter-based LP measurement for each parameter at each time point.

	Sensitivity	Specificity	PPV	NPV		Sensitivity	Specificity	PPV	NPV
Parameter					Time Point				
Worst fQRS	61	67	61	89	0:00	57	74	57	88
Best fQRS	43	80	43	86	4:00	57	75	57	89
Worst RMS40	61	65	61	88	8:00	43	75	43	86
Best RMS40	43	85	43	87	12:00	52	57	52	76
Worst LAS40	65	63	65	87	16:00	61	78	61	90
Best LAS40	43	84	43	87	20:00	57	80	57	90
Mean values of 3 LP parameters	48	78	48	85					

fQRS, filtered QRS duration; LAS40, duration of low-amplitude signals < 40 μV in the filtered QRS complex terminus; LP, late potential; NPV, negative predictive value; PPV, positive predictive value; RMS40, root mean square voltage of the terminal 40 ms in the filtered QRS complex.

**Table 5 medicina-59-01460-t005:** Relationship between LP measurement timing and lethal arrhythmia.

For Each LP Parameter	Univariate			Multivariate			Multivariate (Stepwise)		
	OR	95% CI	*p*	OR	95% CI	*p*	OR	95% CI	*p*
Worst fQRS	3.11	1.22–7.91	<0.001	1.00	0.87–11.56	0.998			
Best fQRS	4.13	1.55–11.03	<0.001						
Worst RMS40	2.85	1.12–7.23	<0.001	0.332	0.021–5.36	0.437			
Best RMS40	4.46	1.66–12.0	<0.001						
Worst LAS40	3.75	1.45–9.71	0.006	10.41	0.58–185.46	0.111	3.75	1.45–9.71	0.006
Best LAS40	4.14	1.55–11.04	<0.001						
Mean values of three LP parameters	3.76	1.45–9.75	<0.001						
For each time point	Univariate			Multivariate			Multivariate (stepwise)		
	OR	95% CI	*p*	OR	95% CI	*p*	OR	95% CI	*p*
0:00	3.61	1.42–9.19	0.007	0.66	0.75–5.81	0.710			
4:00	3.80	1.49–9.70	<0.001	0.93	0.084–10.27	0.953			
8:00	2.97	1.14–7.70	<0.001	0.21	0.024–1.75	0.148			
12:00	4.67	1.80–12.07	<0.001	3.16	0.29–33.91	0.342			
16:00	4.41	1.11–11.36	<0.001	2.74	0.39–19.26	0.310			
20:00	5.00	1.93–13.02	<0.001	4.40	0.52–37.25	0.174	4.89	1.88–12.7	0.001

fQRS, filtered QRS duration; LAS40, duration of low-amplitude signals < 40 μV in the filtered QRS complex terminus; LP, late potential; RMS40, root mean square voltage of the terminal 40 ms in the filtered QRS complex; CI, confidence interval; OR, odds ratio.

**Table 6 medicina-59-01460-t006:** (**A**) Factors influencing diurnal variation in LP parameters (MI-VT group); (**B**) factors influencing diurnal variation in LP parameters (MI-non-VT group); (**C**) factors influencing diurnal variation in LP parameters (control group).

(A)						
fQRS	R = 0.490			R = 0.448 *		
	β	*p*	VIF	β	*p*	VIF
Body position	0.031	0.770	1.527			
log Noise (μV)	0.081	0.484	1.812			
log HR (bpm)	−0.188	0.085	1.599	−0.180	0.037	1.016
log pNN50 (%)	0.256	0.270	7.296	0.433	<0.001	1.016
log RMSSD (ms)	0.212	0.417	9.246			
log ASDNN (ms)	−0.180	0.382	5.771			
log SDANN (ms)	−0.021	0.832	1.325			
log VLF (ms^2^)	0.187	0207	2.977			
log HFnu (TP)	0.183	0.140	2.070			
log LF/HF	0.086	0.400	1.399			
RMS40	R = 0.500			R = 0.305 *		
	β	*p*	VIF	β	*p*	VIF
Body position	−0.092	0.417	1.397			
log Noise (μV)	−0.018	0.881	1.550			
log HR (bpm)	0.422	0.000	1.441	0.305	0.003	1.000
log pNN50 (%)	−0.230	0.336	6.211			
log RMSSD (ms)	−0.066	0.790	6.619			
log ASDNN (ms)	0.180	0.415	5.264			
log SDANN (ms)	0.077	0.509	1.483			
log VLF (ms^2^)	0.076	0.648	2.971			
log HFnu (TP)	0.796	0.002	7.084			
log LF/HF	0.733	0.007	7.566			
LAS40	R = 0.392			R = 0.292 *		
	β	*p*	VIF	β	*p*	VIF
body position	0.013	0.916	1.492			
log Noise (μV)	−0.010	0.942	1.788			
log HR (bpm)	−0.330	0.010	1.524	−0.261	0.011	1.000
log pNN50 (%)	0.081	0.749	6.233			
log RMSSD (ms)	0.148	0.568	6.483			
log ASDNN (ms)	−0.032	0.890	5.196			
log SDANN (ms)	−0.008	0.950	1.475			
log VLF (ms^2^)	−0.134	0.448	2.970			
log HFnu (TP)	−0.525	0.057	7.175			
log LF/HF	−0.402	0.154	7.582			
**(B)**						
**fQRS**	**R = 0.366**			**R = 0.353 ***		
	**β**	** *p* **	**VIF**	**β**	** *p* **	**VIF**
Body position	−0.054	0.348	1.287			
log Noise (μV)	−0.036	0.529	1.308			
log HR (bpm)	−0.021	0.725	1.436			
log pNN50 (%)	0.305	0.001	3.092	0.298	0.001	2.945
log ASDNN (ms)	−0.235	0.028	4.480	−0.222	0.029	4.047
log SDANN (ms)	0.005	0.934	1.406			
log VLF (ms^2^)	−0.184	0.037	3.027	−0.180	0.030	2.684
log HFnu (TP)	−0.038	0.692	3.680			
log LF/HF	0.190	0.071	4.291	0.209	0.002	1.822
RMS40	R = 0.367			R = 0.327 *		
	β	*p*	VIF	β	*p*	VIF
Body position	−0.039	0.493	1.287			
log Noise (μV)	0.155	0.007	1.308	0.156	0.002	1.000
log HR (bpm)	0.046	0.446	1.436			
log pNN50 (%)	−0.241	0.007	3.092	−0.208	0.003	1.903
log ASDNN (ms)	0.136	0.203	4.480	0.206	0.003	1.902
log SDANN (ms)	0.075	0.209	1.406			
log VLF (ms^2^)	0.119	0.175	3.027			
log HFnu (TP)	−0.027	0.777	3.680			
log LF/HF	−0.157	0.134	4.291			
LAS40	R = 0.344			R = 0.314 *		
	β	*p*	VIF	β	*p*	VIF
Body position	0.029	0.617	1.287			
log Noise (μV)	−0.119	0.041	1.308	−0.122	0.017	1.000
log HR (bpm)	−0.008	0.890	1.436			
log pNN50 (%)	0.265	0.003	3.092	0.219	0.002	1.903
log ASDNN (ms)	−0.221	0.041	4.480	−0.224	0.001	1.902
log SDANN (ms)	−0.086	0.154	1.406			
log VLF (ms^2^)	−0.008	0.929	3.027			
log HFnu (TP)	0.070	0.472	3.680			
log LF/HF	0.155	0.142	4.291			
**(C)**						
**fQRS**	**R = 0.458**			**R = 0.452 ***		
	**β**	** *p* **	**VIF**	**β**	** *p* **	**VIF**
Body position	−0.035	0.556	1.352			
log Noise (μV)	−0.473	<0.001	1.271	−0.484	<0.001	1.179
log HR (bpm)	0.139	0.050	1.948	0.141	0.022	1.473
log pNN50 (%)	−0.048	0.631	3.860			
log ASDNN (ms)	0.118	0.332	5.753			
log SDANN (ms)	−0.024	0.705	1.530			
log VLF (ms^2^)	−0.105	0.298	3.985			
log HFnu (TP)	−0.150	0.319	8.789	−0.129	0.028	1.356
log LF/HF	0.004	0.982	9.626			
RMS40	R = 0.396			R = 0.356 *		
	β	*p*	VIF	β	*p*	VIF
Body position	0.112	0.078	1.385			
log Noise (μV)	0.138	0.042	1.588	0.147	0.008	1.049
log HR (bpm)	−0.081	0.265	1.840			
log pNN50 (%)	0.123	0.249	3.925	0.094	0.089	1.049
log ASDNN (ms)	−0.013	0.911	4.489			
log SDANN (ms)	0.035	0.552	1.227			
log VLF (ms^2^)	−0.075	0.407	2.837			
log HFnu (TP)	−0.027	0.768	2.795			
log LF/HF	0.001	0.987	1.523			
LAS40	R = 0.575			R = 0.563 *		
	β	*p*	VIF	β	*p*	VIF
Body position	0.032	0.558	1.352			
log Noise (μV)	−0.633	<0.001	1.271	−0.609	<0.001	1.169
log HR (bpm)	0.240	<0.001	1.948	0.245	<0.001	1.169
log pNN50 (%)	0.100	0.278	3.860			
log ASDNN (ms)	−0.008	0.946	5.753			
log SDANN (ms)	0.035	0.548	1.530			
log VLF (ms^2^)	−0.026	0.781	3.985			
log HFnu (TP)	0.051	0.715	8.789			
log LF/HF	0.152	0.295	9.626			

log ASDNN, logarithm of mean of the standard deviations of all NN intervals for all 5 min segments in 24 h HF; log HR, logarithm of heart rate; log HFnu, logarithm of power in the high-frequency area normalized unit; log LF/HF, logarithm of power in the low-frequency/power in the high-frequency ratio; log pNN50, logarithm of percent of difference between adjacent normal RR intervals greater than 50 ms; log RMSSD, logarithm of root mean square successive difference; log SDANN, logarithm of standard deviation of 5 min average NN intervals; VIF, variance inflation factor; log VLF, logarithm of low-frequency area. *, variables by multiple linear regression with stepwise selection; logarithm of root mean square successive difference (log RMSSD) was removed from analysis because of multicollinearity.

**Table 7 medicina-59-01460-t007:** Values of late potential predicting event after MI.

Author (Published Year)	LP Method	No of Pt	No of Pt AE (%)	Sensitivity	Specificity	PPV	NPV
Strasbert et al. (1993) [23]	Real-time LP	100	12 (12)	50	61	15	90
Denes et al. (1994) [24]	Real-time LP	787	33 (4)	60	98	20	89
Bloomfield et al. (1996) [25]	Real-time LP	177	16 (9)	69	62	15	95
Zimmerman et al. (1997) [26]	Real-time LP	458	32 (6)	44	83	20	94
Ikeda et al. (2000) [2]	Real-time LP	102	15 (14)	53	85	38	91
Amino et al. (2019) [14]	Holter-based LP	90	421 (21)	53	31	NA	NA
Hashimoto et al. (2020) [8]	Holter-based LP	104	11 (10)	63	80	20	95
This study, worst LAS40 (2023)	Holter-based LP	126	23 (18)	65	63	65	87
This study, 20:00 (2023)	Holter-based LP	126	24 (18)	57	80	57	90

AE, adverse event; LP, late potential; MI, myocardial infarction; No, number; NPV, negative predictive value; PPV, positive predictive value; Pt, patient.

## Data Availability

The original contributions presented in this study are included in the manuscript and the Appendix A. Further inquiries can be directed to the corresponding author.

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
