# Peer review of "Diurnal Variation in and Optimal Time to Measure Holter-Based Late Potentials to Predict Lethal Arrhythmia after Myocardial Infarction"

_medicina, 2023, doi:10.3390/medicina59081460_

Round 1

Reviewer 1 Report

Well written paper with appropriate and comprehensive statistics. Please compare and explain the relatively low sensitivity and specificity of present study.

Minor issues to be resolved:

Abstract

Last sentence says „noise levels influence diurnal variability” is not correct, it is just the „MEASURED diurnal variability”.

Introduction

Please detail the sensitivity and specificity reported in the referenced literature [1..5].

Materials and Methods

Table 1 – number/per cent of hypertension in MI-VT group is mistyped.

Line 99: the bandwidth of the ECG is not clear. The bandwidth of each ECG record was set at 40-250Hz by H-ECG system? The Ag/AgCl electrodes should be mentioned at the beginning of the paragraph since this is the first member of the signal chain.

Line122: the < may be in wrong direction at the RMS40.

Fig 1 is difficult to see, please magnify it and reorganize it in two columns and three rows.

Line143: pNN50 correctly

Results

Line196: left anterior descending correctly

The tables will be reformatted to fit one page.

In 3.3 there is a discussion, please transfer it to the Discussion part.

Discussion

It is difficult to digest 4.1-4.2., please reorganize or rephrase. However, the Reader cannot see possible explanations of the findings: e.g. why are there differences in influencing factors among the groups? In 4.3. there can be other clinical consequences besides using AI or deep learning methods in LP analysis.

Conclusions

The HR should be considered separately from the noise level. Noise levels above a certain limit should preclude LP analysis.

Author Response

Title: Diurnal variation of and optimal time to measure Holter-based late potentials to predict lethal arrhythmia after myocardial infarction

Dear Editors,

We sincerely thank the reviewers for considering our manuscript and for their valuable comments. We have quoted those comments below in blue, with our responses in black. We revised the manuscript in each instance to address the reviewers’ comments. The changes in the manuscript text are highlighted in red. In addition, the entire manuscript has been rechecked and any necessary changes made for correctness and consistency.

We hope you find that our revised manuscript meets the publication requirements of Medicina, and look forward to hearing the decision regarding the revised manuscript.

Yours sincerely,

Kenichi Hashimoto
Department of General Medicine, National Defense Medical College
3-2 Namiki, Tokorozawa City, Saitama, Japan, 359-8513
Phone: +81-42-991-1211 (ex. 3633)
Fax: +81-42-995-1238
Email: [email protected]

Reviewer 1

Comments and Suggestions for Authors

Comment Well written paper with appropriate and comprehensive statistics. Please compare and explain the relatively low sensitivity and specificity of present study.

Response Thank you for your kind comments. Please see our responses below regarding our explanation of the sensitivity and specificity of the present study and comparison to those of other studies.

Minor issues to be resolved:

Abstract

Comment Q1. Last sentence says „noise levels influence diurnal variability” is not correct, it is just the „MEASURED diurnal variability”.

Response Thank you for your suggestion. We have addressed the above issue. The description has been corrected to read “Changes in autonomic nervous system activity, including heart rate, were factors influencing diurnal variability. Increased parasympathetic activity or bradycardia may exacerbate Holter-based LP parameters.” (Lines 34-36)

Introduction

Comment Q2. Please detail the sensitivity and specificity reported in the referenced literature [1..5].

Response Thank you for your suggestion. We have researched the previous studies and checked the sensitivity and specificity of both real-time LPs and Holter-based LPs in patients with post-MI status. The reviewer’s suggestion is to detail the sensitivity and specificity reported in the referenced literature [1–5]. In other words, the suggestion is to research sensitivity and specificity regarding patients with not only MI but also DCM, ARVC, and sarcoidosis. Considering this matter carefully, we thought it better to limit this analysis to the literature regarding post-MI patients. Therefore, we added references 23–26 and added a comparison of LP test accuracy between previous studies and our study to the discussion in lines 338-349. In addition, we have added Table 7 to provide a clear presentation of the findings of the previous studies and our study.

The added text is as follows:

Table 7 shows the predictive values of late potential events after MI as reported in the previous literature [23-27]. Generally, LP testing is characterized by high NPVs and low PPVs. Representative real-time LPs in previous studies yielded PPVs of 15–38% and NPVs of 89–91%. Of note, the PPVs found in our study (57 and 65%) are higher than those in previous studies, including those using standard real time LP recording (Table 7). This may represent an advantage of Holter-based LPs, provided that the measurement point of LP determination is optimal. In contrast, the sensitivity is equivalent but the specificity and NPV are relatively low in our study compared to those in the previous studies. We speculate that the reason for this phenomenon is that the proportion of patients with cardiac events relative to the total was higher in this study than that in other studies (18%). Therefore, the low number of LP (-), VT (-) patients may have led to the somewhat lower specificity and NPV.

(Line 334-345).

Materials and Methods

Comment Q3. Table 1 – number/per cent of hypertension in MI-VT group is mistyped.

Response: Thank you for this correction. We have revised the mistyped figure from “23%” to “78%”.

Comment Q4. Line 99: the bandwidth of the ECG is not clear. The bandwidth of each ECG record was set at 40-250Hz by H-ECG system? The Ag/AgCl electrodes should be mentioned at the beginning of the paragraph since this is the first member of the signal chain.

Response We have moved the description regarding Ag/AgCl electrodes — “Orthogonal X, Y, and Z bipolar leads with silver-silver chloride electrodes (Blue SENSOR®; METS, Tokyo, Japan) were used for all LP recordings” — to the beginning of the paragraph (Lines 99-100). We have also carefully confirmed the information regarding bandwidth of each ECG record. It is correct that the bandwidth of each ECG record was set at 40-250 Hz. Please see page 12 of the attached operating instructions file from the manufacture (Ela Medical, Inc.). Further, in references 6 and 8, the description “ECG record was set at 40-250 Hz by the Ela®H-ECG system” was mentioned. Therefore, we added references 6 and 8 to Line 103.

Comment Q5. Line122: the < may be in wrong direction at the RMS40.

Response Thank you for your suggestion. We have checked this carefully and we believe this description to be correct (Line 126). Please see page 12 of the attached operating instructions file from the manufacturer (Ela Medical, Inc.) .

Comment Q6. Fig 1 is difficult to see, please magnify it and reorganize it in two columns and three rows.

Response Thank you for your valuable suggestion. We have magnified and reorganized Figure 1, and now it is easier to read. However, because the figure contains a total of nine panels (three for each three parameters), it is not easily organized into two columns and three rows. We will gladly reformat the figure further if the present format is not adequate.

Comment Q7. Line143: pNN50 correctly

Response We have corrected the word “PNN50” to “pNN50”. All additional instances of this error have been corrected. (Lines 32, 148, 179, 269, 395, and Table 6.A, B and C)

Results

Comment Q8. Line196: left anterior descending correctly

Response Thank you for your correction. We have corrected the phrase from “left atrial descending” to “left anterior descending.” (Line 200)

Comment Q9. The tables will be reformatted to fit one page.

Response Thank you for your valuable comment. We have reformatted each of the tables to fit one page, using the minimum 8 pt font size where necessary.

Comment Q10. In 3.3 there is a discussion, please transfer it to the Discussion part.

Response We have transferred a portion of section 3.3 of the Results, “Factors Influencing Diurnal Variability of Holter-Based LPs,” to the Discussion section (Line 376-387).

Discussion

Comment Q11. It is difficult to digest 4.1-4.2., please reorganize or rephrase.

Response Thank you very much for your valuable suggestion. We have reorganized the Discussion section to be consistent with the Results section. Specifically, the headings of the Discussion section are as follows: 4.1 Diurnal Variation of Holter-Based LP; 4.2 Optimal LP Measurement Timing for Predicting VT; 4.3 Factors Influencing Holter-Based LP Values; 4.4 Clinical Implications; and 4.5 Limitations. (Lines 275-442)

Comment Q12. However, the Reader cannot see possible explanations of the findings: e.g., why are there differences in influencing factors among the groups?

Response Thank you very much for your important suggestion. As you pointed out, there was a slight difference in the results among the MI-VT, MI-non-VT, and control groups. It was challenging to solve this problem clearly, and the following measures were taken. First of all, noise is a major problem in Holter-LP, and it affects the actual measured value of LP. As you pointed out earlier, noise affects measured diurnal variation, but it cannot be a factor in physiological diurnal variation. Therefore, we removed noise from the analysis and redid all of the multivariate analyses. However, the resulting model coefficient R of the multivariate analysis was much lower than shown in the existing Table 6, in the range of R=0.1–0.3, and we found this not to be feasible as a realistic multivariate model. In addition, the pattern of all possible clinically important factor combinations also showed that the model presented in Table 6 in the initial submission still had the highest model coefficient R.

It is difficult to explain why there were slight differences among the MI-VT, MI-non-VT, and control groups. However, we speculate as follows (added text):

In the multivariate analysis, noise significantly affected LP parameters in the MI-non-VT and control groups, but not in the MI-VT group. Although it is difficult to explain the substantive reason for this difference, it is possible that the MI-VT group results were less affected by noise because of their significantly lower EF and greater LVDD compared to the MI-non-VT and control groups. Though HR did not survive in the multivariate analysis as a significant factor in the MI-non-VT group, HR is normally influenced by the autonomic nervous system. In the MI-non-VT group, log HR was mildly correlated with log pNN50 and log ASDNN, both at R=0.3 (Pearson correlation coefficient), p<0.001. Thus, in all groups, increasing parasympathetic tone or bradycardia may lead to an exacerbation of LP parameters. (Lines 388-398)

Comment Q13. In 4.3. there can be other clinical consequences besides using AI or deep learning methods in LP analysis.

Response Thank you very much for your suggestion. We have added the new section, 4.4 Clinical implications, to reflect other clinical consequences besides using AI or deep learning methods in LP analysis. The description is as follows:

Holter-based LP can be performed simultaneously with routine Holter ECG, saving time for both patients and medical stuffs compared to the conventional real time LP measurement. Furthermore, according to the present study result, the likelihood of missing the good accuracy test result of Holter based LP would be greatly decreased by reading mainly at the time worst value of LAS40 or around 20:00. (Lines 429-433)

Conclusions

Comment Q14. The HR should be considered separately from the noise level. Noise levels above a certain limit should preclude LP analysis.

Response Thank you for your suggestion. The statement regarding noise level has been separated from the HR and HRV analysis. We have revised the Conclusions section as follows:

The involvement of the autonomic nervous system, reflected by HR, has been suggested as a factor influencing both patients with post-MI status and healthy control participants. Increasing parasympathetic activity or bradycardia tend to exacerbate Holter-based LP parameters. (Lines 448–451)

The parts of the document that have been changed or repositioned are marked in red. Items in the revised manuscript that have been identified by Peer Review 1

are marked in yellow.

Finally, this is a retrospective clinical study that was approved by the Ethics Committee prior to its start and disclosed on the hospital's website. Therefore, we have deleted the sentence "Written informed consent was obtained from all patients" from Lines 91-92, because written consent was not obtained from each participant. We apologize sincerely for this.

Reviewer 2 Report

The design and writing of the work is interesting and there are a few points that need to be addressed:

- The material and method section is well written, but there are a few problems in the discussion section:

- You should include the main findings of the study in the first part of the discussion.

-The discussion part is short. It is more necessary to compare your own findings with the literature in the discussion part.

- In the Discussion section, you should have included "Ambulatory blood pressure results and heart rate variability in patients with premature ventricular contractions. Clinical and Experimental Hypertension 40 (3), 251-256, 2018."

Author Response

Title: Diurnal variation of and optimal time to measure Holter-based late potentials to predict lethal arrhythmia after myocardial infarction

Dear Editors,

We sincerely thank the reviewers for considering our manuscript and for their valuable comments. We have quoted those comments below in blue, with our responses in black. We revised the manuscript in each instance to address the reviewers’ comments. The changes in the manuscript text are highlighted in red. In addition, the entire manuscript has been rechecked and any necessary changes made for correctness and consistency.

We hope you find that our revised manuscript meets the publication requirements of Medicina, and look forward to hearing the decision regarding the revised manuscript.

Yours sincerely,

Kenichi Hashimoto
Department of General Medicine, National Defense Medical College
3-2 Namiki, Tokorozawa City, Saitama, Japan, 359-8513
Phone: +81-42-991-1211 (ex. 3633)
Fax: +81-42-995-1238
Email: [email protected]

Reviewer 2

Comments and Suggestions for Authors

The design and writing of the work is interesting and there are a few points that need to be addressed:

- The material and method section is well written, but there are a few problems in the discussion section:

Comment Q1.- You should include the main findings of the study in the first part of the discussion.

Response Thank you very much for your suggestion. We have added a statement of the main findings of the study to the first part of the discussion as “This study found that Holter-based LPs and their associated factors had significant diurnal variation in the MI-VT, MI-non-VT, and control groups (Tables 2, 3). The optimal times for measuring Holter-based LPs to predict VT were the time of the worst LAS40 reading and nighttime (20:00) (Figure 2, Tables 4, 5). The factors that influenced LP diurnal variation were aspects of autonomic nerve activity such as log pNN50 (%), logHFnu, log ASDNN or HR (Table 6). On the other hands, noise is the factor which influence to measured diurnal variation of LPs.”. (Lines 275–281)

Comment Q2.-The discussion part is short. It is more necessary to compare your own findings with the literature in the discussion part.

Response Thank you very much for your valuable suggestion, with which we are in complete agreement. Accordingly, we have described our own findings and the literature separately in the Discussion section, and have compared and discussed them as follows:

4.1. Diurnal variation of Holter based LP

Our own findings: Lines 295–300

Compared with the literature and discussed: Lines 300-315

4.2. Optimal Measurement Timing of LP for predicting VT

Our own findings: Lines 317-322

Compared with the literature and discussed: Lines 322-332

4.3. Factors Influencing Holter based LP Values

Our own findings: Lines 376-387

Compared with the literature and discussed: Lines 387-398

This has enhanced the content of the discussion, expanding it from 992 to 1718 words.

Comment Q3.- In the Discussion section, you should have included "Ambulatory blood pressure results and heart rate variability in patients with premature ventricular contractions. Clinical and Experimental Hypertension 40 (3), 251-256, 2018."

Response Thank you very much for your kind suggestion. We have cited the above reference and discussed it relative to our own findings in the Discussion section as follows:

Askin et al. [27] examined the characteristics of HRV in participants with no apparent cardiac disease divided into groups with more and fewer PVCs. They found that LF/HF was higher in the high PVC group than in the low PVC group, both at night and during the day, and that higher LF/HF was a risk factor for increased PVCs in multivariate analysis. Although the LF/HF ratio is often interpreted as a balance between sympathetic and parasympathetic activity, the authors hypothesize that sympathetic hyperactivity rather than altered parasympathetic activity underlies the PVC increase. PVCs and nonsustained VT are important because they are key triggers in the generation of lethal arrhythmias [28]. Fortunately, in our study, parasympathetic tone resulted in increased arrhythmogenicity, the opposite of the autonomic modulation underlying increased PVCs. (Lines 400-409).

The parts of the document that have been changed or repositioned are marked in red. Items in the revised manuscript that have been identified by Peer Review 2 are marked in green.

Finally, this is a retrospective clinical study that was approved by the Ethics Committee prior to its start and disclosed on the hospital's website. Therefore, we have deleted the sentence "Written informed consent was obtained from all patients" from Lines 91-92, because written consent was not obtained from each participant. We apologize sincerely for this.

Round 2

Reviewer 2 Report

Successful corrections. No comments